# An ELISA Using Synthetic Mycolic Acid-Based Antigens with DIVA Potential for Diagnosing Johne’s Disease in Cattle

**DOI:** 10.3390/ani14060848

**Published:** 2024-03-09

**Authors:** Paul S. Mason, Thomas Holder, Natasha Robinson, Brendan Smith, Rwoa’a T. Hameed, Juma’a R. Al Dulayymi, Valerie Hughes, Karen Stevenson, Gareth J. Jones, H. Martin Vordermeier, Shawn Mc Kenna, Mark S. Baird

**Affiliations:** 1Diagnostig Ltd., M-SParc, Gaerwen, Anglesey LL60 6AG, Wales, UK; mason@diagnostig.com; 2Animal and Plant Health Agency, Addlestone KT15 3NB, Surrey, UK; thomas.holder@apha.gov.uk (T.H.); gareth.jones@apha.gov.uk (G.J.J.); martin.vordermeier@apha.gov.uk (H.M.V.); 3Department of Health Management, Atlantic Veterinary College, Charlottetown, PE C1A 4P3, Canada; ngrobinson@upei.ca (N.R.); slmckenna@upei.ca (S.M.K.); 4Bimeda Animal Health Ltd., Airton Close, Airton Road, Tallaght, D24 FH9V Dublin, Ireland; bsmith@bimeda.com; 5School of Natural and Environmental Sciences, Bangor University, Bangor LL57 2UW, Wales, UK; rwoaatareq@uomosul.edu.iq (R.T.H.); jumaaldulayymi@gmail.com (J.R.A.D.); 6College of Science, University of Mosul, Mosul 41002, Iraq; 7Moredun Research Institute, Penicuik EH26 0PZ, Midlothian, UK; val.hughes@moredun.ac.uk (V.H.);

**Keywords:** MAP, mycolic acids, serodiagnosis, TDM, TMM, GMM, DIVA, lipid ELISA, *Mycobacterium avium* subspecies *paratuberculosis*, Johne’s disease, vaccine

## Abstract

**Simple Summary:**

Johne’s disease, caused by *Mycobacterium avium* subsp. *paratuberculosis* (MAP), is endemic in cattle herds in the UK and throughout the world. It is a wasting disease with a protracted incubation period, during which clinical signs are not apparent, and is responsible for considerable economic losses. Infection can be transmitted silently from animal to animal, and MAP can find its way into the food chain through the products of infected animals. Ante-mortem diagnosis of the disease is normally achieved through detection of the organism (by faecal culture or PCR) or serological tests, but agreement as to a positive diagnosis is often poor and the certainty of identifying all positive animals is low, particularly in early-stage infections. Mycobacterial cells contain high-molecular-weight lipids, mycolic acids (MAs), which are recognised by antibodies in active infections. Here, we show that a simple assay using different classes of single synthetic MAs can detect antibodies in serum from cattle with MAP infections and distinguish uninfected animals from animals positive both to faecal culture and a commercial serodiagnostic assay. It does so without interference from vaccination either for MAP (a Gudair vaccination) or for bovine TB (BCG vaccination). When the assay is optimised to distinguish negative samples from those positive only by faecal PCR, it identifies additional positives that are not seen using a commercial serodiagnostic assay; this suggests that, with further development, it may provide a method for detecting infections before the commercial serodiagnostic assay can.

**Abstract:**

The problem: Ante-mortem diagnosis of Johne’s disease, caused by *Mycobacterium avium* subsp. *paratuberculosis* (MAP), is normally achieved through faecal culture, PCR, or serological tests, but agreement as to which samples are positive for Johne’s disease is often poor and sensitivities are low, particularly in early-stage infections. The potential solution: Mycobacterial cells contain very complex characteristic mixtures of mycolic acid derivatives that elicit antibodies during infection; this has been used to detect infections in humans. Here, we explore its application in providing an assay differentiating infected from vaccinated animals (DIVA assay) for Johne’s disease in cattle. Method: Antibody responses to different classes of mycolic acid derivatives were measured using ELISA for serum from cattle positive for MAP by both faecal PCR and commercial serum ELISA, or just by PCR, and from animals from herds with no history of Johne’s disease, bovine tuberculosis reactors, BCG-vaccinated, BCG-vaccinated and *M. bovis*-infected, and Gudair-vaccinated animals. Results: The best-performing antigens, ZAM295 and ST123—the latter a molecule present in the cells of MAP but not of *Mycobacterium bovis*—achieved a sensitivity of 75% and 62.5%, respectively, for serum from animals positive by both faecal PCR and a commercial MAP serum ELISA, at a specificity of 94% compared to 80 no-history negatives. Combining the results of separate assays with two antigens (ST123 and JRRR121) increased the sensitivity/specificity to 75/97.5%. At the same cut-offs, animals vaccinated with Gudair or BCG vaccines and bTB reactors showed a similar specificity. The specificity in BCG-vaccinated but *M. bovis*-infected animals dropped to 85%. Combining the results of two antigens gave a sensitivity/specificity of 37.5/97.5% for the full set of 80 PCR-positive samples, detecting 30 positives compared 16 for IDEXX. Conclusion: Serum ELISA using synthetic lipids distinguishes effectively between MAP-negative cattle samples and those positive by both PCR and a commercial MAP serodiagnostic, without interference by Gudair or BCG vaccination. It identified almost twice as many PCR positives as the commercial serodiagnostic, offering the possibility of earlier detection of infection.

## 1. Introduction 

*Mycobacterium avium* subsp. *paratuberculosis* (MAP) causes disease in ruminants, camelids, and leporids and can infect a wide range of animals [1,2,3,4,5,6]. MAP causes Johne’s disease in cattle, characterised by weight loss and diarrhoea and is associated with lower economic yields in infected animals. The organism is present in the milk of infected animals and may be incompletely inactivated by pasteurisation. There appear to be increased instances of the presence of MAP in patients with Crohn’s disease, although causality has not been proven. Although Johne’s disease is not reportable in Great Britain and many other countries, it is notifiable in some countries and in certain species. There is growing pressure to ensure food products are from ‘Johne’s-free’ animals [1]. 

The organism is primarily transferred via ingestion of infected milk or food and water contaminated with faeces from infected animals. It can survive for long periods in the environment, 6–18 months in water and 5–8 months on pasture [7]. Not all animals exposed to MAP become infected or go on to develop disease. The disease develops slowly in cattle and there are a number of models of disease development. In one, three stages are described: (i) subclinical infection, defined as MAP infection without demonstrable pathology in tissues; (ii) subclinical disease, defined as the presence of pathology without weight loss or diarrhoea; (iii) clinical disease, defined as the presence of pathology with weight loss and/or diarrhoea [7]. Gross pathology, and more specifically histopathology, provides the most definitive diagnosis of the disease stages defined above, and since it can only be determined post-mortem, differentiating subclinical infection and subclinical disease using ante-mortem tests is not possible. Thus, there are no routine, reliable methods for the ante-mortem diagnosis of early-stage infection [8], and later stages of disease are normally diagnosed by either faecal culture or PCR, or a serodiagnostic assay detecting disease antibodies. Elevated MAP antigen induced interferon (IFN)-γ production in sub-clinically infected animals has been used in diagnosis [9], but this cytokine declines in the clinical stage [10].

Detection of the organism in faeces by culture or PCR is often used as evidence of infection. However, while it is certainly evidence of exposure, it may not always be indicative of infection but rather the result of the passage of contaminated material through the gut (passive shedding) [11]. Furthermore, infected animals may exhibit intermittent shedding and may return to a more active phase of disease with frequent shedding, possibly triggered by stress during lactation or parturition. Whatever the trigger, it has been suggested that the progression from subclinical to clinical disease coincides with a shift in immune function from a Th1 to a Th2 response. Because there are no other widely used, reliable methods for the detection of early infection, the balance of these possibilities remains unclear. There is therefore often a poor correlation between faecal culture or PCR and a serodiagnostic assay, particularly in early-stage disease [7,8,12,13]. The current widely used commercial serodiagnostic assays generally provide excellent specificity, but recommended cut-offs often lead to very low sensitivities, certainly compared to culture or PCR. Moreover, specificity may be low in geographical regions where animals are exposed to environmental mycobacteria [14]. Although predictive models have been developed [15,16], the use of faecal PCR on its own must also be treated with caution, and the recommended maximum number of quantification cycles that can safely be interpreted as a positive diagnosis was reduced in one recent study from <37 to <33, with any value from 33 to 40 potentially being a false positive and requiring supporting evidence [17]. It is worth noting that false results could be caused by inhomogeneity of sampling, by passive shedding, by intermittent shedding, or by changes in PCR quantification cycles when appropriate controls are not applied.

A major challenge in controlling Johne’s disease is therefore the ability to detect infected cattle prior to appearance of disease signs, such as diarrhoea and heavy faecal shedding of MAP. Cattle that appear healthy may not be routinely evaluated using serial test bleeds and analysis unless they are part of a cattle health scheme. An assay that detects responses in blood or serum at an earlier point in disease development, and one that can help to determine whether a greater number of positive PCR do indeed correspond to disease and not exposure and direct pass-through, is therefore highly desirable [18]. 

A further complication is the need for diagnostic methods the results of which are not affected by vaccination for MAP [19,20], or by developing vaccines for bovine TB [21], and indeed by the statutory skin test for bovine tuberculosis (bTB).

Like other mycobacteria, the cells of MAP contain complex mixtures of very long chain hydroxy-fatty acids, mycolic acids (MA) (Figure 1), and of their sugar esters, in particular trehalose mono- and di-mycolates (TMM and TDM) [22]. The natural mixtures comprise molecules with a variety of groups X and Y and chain lengths a–d. The detailed pattern of these molecules forms a fingerprint which can be used directly to identify a particular mycobacterium, for example by mass spectrometry. In *Mycobacterium tuberculosis* and *Mycobacterium bovis *(*M. bovis* throughout this manuscript), the MAs are largely from three classes, alpha (X includes a cyclopropane), keto (X includes a ketone with an adjacent methyl branch), and methoxy (X includes a methoxy group with an adjacent methyl branch); commonly, group Y includes a cyclopropane fragment. In *Mycobacterium avium* complex (MAC), the keto class is replaced by ‘wax esters’ containing an additional oxygen in group X [23]. Other non-tuberculous mycobacteria (NTM) often contain much simpler groups X and Y containing either *cis*- or *trans*-alkenes. Within an infection such as tuberculosis (TB), the balance of specific MA classes (and the stereochemistry within the classes) is known to change with disease progression [22].

Mycolic acid and mycolate ester extracts of cells of *Mycobacterium tuberculosis* are antigenic to antibodies in the serum of patients with active tuberculosis. They have therefore been evaluated in the diagnosis of the disease, using ELISA [24,25,26]. In the same way, complex mixtures of trehalose mycolates extracted from the cells of *Mycobacterium avium* can be used to distinguish MAC infection from MTb infection [27]. Single synthetic mycolic acids and their sugar esters have also been used in TB diagnosis; in these studies, a multiplex assay was used in which the results of individual assays with several antigens were combined statistically to optimise sensitivity and specificity [28]. A similar method has been used in the diagnosis of infections by *Mycobacterium abscessus* in immune-compromised patients [29]. 

We now report the evaluation of unique synthetic mycolic acids and their sugar esters as antigens in ELISA for the detection of antibodies generated to these lipids in MAP infection in cattle, including control animals, and those that had been vaccinated for bTB or for MAP. A key aim was to determine whether specific antigens or antigen combinations would be recognised by antibodies to mycolic acid-based lipids, and if so, whether the responses observed could be correlated with some PCR data better than those to standard antigens in currently used serodiagnostics. This might then allow for the confirmation of developing disease at an earlier time and represent the initial stage of the rigorous process of evaluation of an improved diagnostic assay [30]. Early diagnosis has also been the target for serodiagnostics using arrays of protein antigens [31].

## 2. Materials and Methods

### 2.1. Serum Samples

Cattle serum samples were provided either from Atlantic Veterinary College, Prince Edward Island (PEI) in Canada or from Moredun Research Institute (MRI) or Animal and Plant Health Agency (APHA) in the UK.

#### 2.1.1. Positive Serum Samples

Faecal PCR-positive serum samples (n = 80) were provided by PEI. Serum samples were from animals from MAP-infected herds. Animals were considered infected with MAP if they tested positive by faecal PCR and/or a commercial paraTB ELISA manufactured by IDEXX. Of 80 animals, 16 were positive by both faecal PCR and IDEXX paraTB ELISA (PEI++ cohort). The remaining 64 samples were faecal PCR-positive but IDEXX ELISA-negative (PEI+− cohort). In this study, the whole herd was first screened by faecal PCR and positive samples were then tested by IDEXX ELISA so no samples in this set were IDEXX ELISA-positive and PCR negative. The PCR quantification cycles for all these samples were within the limits set by the commercial assay used (see below) and these and the IDEXX ELISA results are presented in Appendix A. 

#### 2.1.2. Negative Serum Samples

(i)From Prince Edward Island (PEI) (n = 80)

Two sets of 40 sera from negative animals from two farms (F5 and F6) were provided by PEI. Both farms had participated in annual herd testing for at least 10 years with no history of infection. These animals were faecal PCR-negative and IDEXX ELISA-negative (PEI−−). Animals from herd F5 were kept outdoors on pasture during suitable times of the year; those from herd F6 were housed exclusively indoors year-round. The PCR and IDEXX ELISA results for all the negative samples are presented in Appendix A.

(ii)From Moredun Research Institute (MRI) (n = 9)

A further set of negative sera were provided by MRI. The animals providing the serum samples were from herds participating in the national paratuberculosis control programme (CHECS-Technical-document-Johnes-disease-rule-change-released-220621-date-01-11-2021.pdf, accessed on 28 December 2023) [32] and the herd had consistently tested negative for MAP by serological testing using ID Screen Paratuberculosis Indirect ELISA (Innovative Diagnostics, Grabels France) for at least the previous 3 years. The farm had no history of bTB and is located in Scotland, which is designated an ‘Officially bovine Tuberculosis-Free’ (OTF) region.

(iii)From Animal and Public Health Authority (APHA) (n = 20)

Five sets of samples were provided by APHA from field animals from a number of farms:(a)Serum Samples from Skin-Test (SICCT)-Negative Animals (n = 20)

Serum samples from 20 SICCT-negative animals were provided by APHA (Control 1–20). Serum samples were taken 10 days post skin test.

(b)Serum samples from Skin-Test (SICCT)-Positive Animals (bTB reactors) (n = 20)

Serum samples from Skin-Test (SICCT)-Positive Animals were provided by APHA (Reactor 1–20). These animals had a natural *M. bovis* infection, which was subsequently confirmed by both the commercial Interferon-gamma Release Assay (IGRA) and post-mortem examination. Serum samples were taken 8 weeks after SICCT.

(c)Serum Samples from BCG-Vaccinated Animals (n = 20)

Serum samples from BCG-vaccinated (but not SICCT-tested) animals were provided by APHA (BCG Vac 1–20). Animals were inoculated with 1–4 × 10^6^ CFU of BCG SSI via the subcutaneous route. Serum samples used in this study were taken 6 weeks post vaccination.

(d)Serum Samples from Gudair-Vaccinated Animals (n = 20)

Serum samples from Gudair-vaccinated animals were provided by APHA (Gudair 1–20). Animals received 2 ml Gudair vaccine (CZ Veterinaria, Spain) via the subcutaneous route. Serum samples were taken 8 weeks post vaccination.

(e)Serum Samples from BCG-Vaccinated, *M. bovis*-infected Animals (n = 20)

Serum samples from BCG-vaccinated, *M. bovis*-infected animals were provided by APHA (BCG Vac-Inf 1–20). These animals were BCG-vaccinated as above, and then experimentally infected 10 weeks post vaccination with *M. bovis* strain AF2122/97. Serum samples used in this study were taken 22 weeks post vaccination.

(iv)Longitudinal negatives from APHA: 90 serum samples from 10 IDEXX Para TBserum ELISA negative and *M. bovis*-free animals over 9 time points. 

These animals had been imported from a bTB-free country at the age of 35–50 days. Serial serum samples were taken at nine timepoints over a period of 47 weeks. Serum samples tested consistently negative by IDEXX ParaTB serum ELISA (Appendix A) and within assay limits by Bovigam IGRA for *M. bovis* infection, though the PPDA values for this assay did show some increased values (Appendix A).

### 2.2. Antigens 

Six antigens were evaluated in this work. Their structures are presented in Appendix A. Antigen MOD171 is based on a cell-wall-bound mycolic acid fragment, a triarabinose dimycolate, incorporating a major α-mycolic acid present in many mycobacteria including MAC [33]. ZAM295 is a trehalose dimycolate of a diene-mycolic acid, a sub-group commonly present in NTM [34]. ST123 is a trehalose dimycolate of a wax ester; this class of molecule is present in MAC, replacing the keto-mycolic acid class present in many other mycobacteria [22,35]. SMP70 is a glucose monomycolate of a common methoxymycolic acid [36]. RT237F2 is an ester of a major α-mycolic acid with an amino-sugar; it is not a known component of mycobacteria, but a simpler analogue of a mycolyl muramyl dipeptide [37]. JRRR121 is a free methoxy-mycolic acid, incorporating a *trans*-cyclopropane fragment; such *trans*-cyclopropanes are major components of the methoxy-mycolic acid class present in MAC [38].

### 2.3. Faecal PCR

DNA was extracted from 2 g of frozen faecal sample using the MAP Extraction System TC-9014-100 (Tetracore, Rockville, MD, USA) according to manufacturer’s instructions. Real-time PCR was carried out using the VetAlert Johne’s Real-Time PCR kit TC-9828-100 (Tetracore, Rockville, MD, USA) in accordance with manufacturer instructions. The qPCR was performed in a Cepheid SmartCycler II Thermocycler (Cepheid, Sunnyvale, CA, USA). After an enzyme activation step (95 °C), a 2-step cycling reaction (95 °C and 62 °C) was used. The cut-off value for the positive control was set to between 20 and 26 cycle threshold values. Samples that were positive within 42 cycles were considered MAP positive as recommended in this assay. Of the 16 PEI++ samples (positive to both PCR and IDEXX), 12 gave quantification cycles below 31, and the highest count was 36.2; none of the 64 PEI+− samples (positive to PCR, negative in IDEXX) gave quantification cycles below 33.

### 2.4. IDEXX ELISA

The IDEXX MAP Ab ELISA test was performed according to the manufacturer’s instructions using a recommended cut-off of 0.55 for a positive.

### 2.5. Lipid ELISA

Responses in ELISA were measured using six individual synthetic lipid antigens, comprising different classes of mycolic acid and sugar ester, as described in Section 2.2. 

The ELISA was carried out on 96-well microplates and the purified antigens were dissolved in THF/n-hexane (1:50, 5 mL) at a concentration of 5.6 μM; 50 μL of this antigen solution was placed in each well. The plates were left to dry at room temperature overnight. In order to reduce non-specific hybridisations, the plates were blocked with casein/PBS containing 0.5% *w*/*v* casein (300 μL/well) and left to incubate at 25 °C for 30 min. The solution was aspirated using the LT-3500 plate washer and the plates flicked dry. The serum (1:40 dilution in casein/PBS) was added to the plate (50 μL/well) and left to incubate at 25 °C for 1 h. The serum was aspirated and the plates washed three times with casein/PBS (400 μL/well) using the same plate washer and the plates were again flicked dry. The secondary antibody (Peroxidase-AffiniPure Goat Anti-Bovine IgG, Stratech Fc; 1:1000 dilution in casein/PBS) was added to the plate (50 μL/well) and left to incubate at 25 °C for 30 min. The plates were washed three times again with casein/PBS (400 μL/well) and dried. OPD substrate (0.2 mg/mL) was added and the plates (50 μL/well) incubated at 25 °C for 30 min; the reaction was then stopped by the addition of 2.5 M H_2_SO_4_ (50 μL/well) and the absorbances of each well were read straight away at 492, 450, and 620 nm using an LT-4000 plate reader; the values at 492 nm were used for the current analysis.

Typically, two replicate samples were analysed. The average optical densities (492 nm) of the replicates and their standard deviations were recorded. On each plate, a common pooled positive control (using samples from animals positive to both faecal PCR and IDEXX ELISA) and a PBS/casein negative control were run alongside test serum samples. Whenever a pooled positive control had to be updated, it was standardised against the existing control. It was also compared to a commercial positive standard obtained from GD Animal Health. The positive control was used to normalise optical density reading on different plates by converting OD readings for individual wells into a percentage of the pooled positive response. Average optical densities were converted into percentage of the positive control for each plate and are recorded in Appendix A.

### 2.6. Statistical Analyses

The current data were analysed in two ways. In the first, the PEI++ samples were used as the positive set as those best corroborated as positive using current assays. In the second, the data were analysed to provide the best fit to the complete set of 80 PCR positive samples.

The significance of differences between the data for each cohort of samples compared either to the PEI++ or PEI+ samples was determined using the Tukey test. In each case, optimal cut-offs for each antigen were initially determined by ROC analysis using the 80 F5 and F6 as negatives. These cut-offs were then applied to analyse the other sets of samples. With some antigens, this gave a low specificity unsuitable for Johne’s diagnosis; in these cases, cut-offs were determined to provide the maximum accuracy (average of sensitivity and specificity) for positive/negative distinction, except that a minimum specificity of 93% was applied, as described in Table 1. 

In each case, the results for various combinations of individual antigens were combined using the R statistics package, identifying the serum samples as known negative or positive sets and generating the best statistical +/− separation. This separated the results along two axes, resulting in a two-dimensional scatter plot. It also provided a prediction of negative or positive status of each sample on a scale from 0 (negative) to 1 (positive). The optimal cut-off for this predicted status was again determined as that giving the maximum accuracy, but with a minimum 93% specificity (Table 1). The resulting combination of responses to various antigens, and the cut-offs established were then applied to predict the status of any other sample or sample set. Box-and-whisker plots were produced in the R statistics package.

## 3. Results

The ELISA results for all the samples with each of the six antigens are presented as % positive control in Appendix A. 

### 3.1. Distinguishing PEI PCR- and IDEXX-Positive Samples and MAP Negatives

Initial analysis was carried out comparing the responses for the 16 PEI++ serum samples with those for the 80 PEI ‘no-history’ negative serum samples (F5 plus F6). Table 1 shows the medians for the PEI++ sample set, together with those for the combined and separate F5 and F6 PEI−− sets with each antigen and the significance values for the differences between the sets, together with optimal combinations of specificity and sensitivity. The final five columns show the corresponding results when the data for individual serum samples and defined antigens were combined using the R statistics package, optimising the positive/negative separation for the 16 PEI++ sera compared to all 80 negatives. Further information, including ROC analysis for each antigen or antigen combination, is provided in Appendix A and Appendix A. The distributions of responses for each cohort of cattle are shown as box-and-whisker plots in Appendix A. 

Antigens MOD171, a model mycobacterial cell-wall fragment, and RT237F2, a model amino-sugar mycolate, gave only a very low sensitivity at >90% specificity. Compared with the complete set of 80 PEI−−-negative sera, the free mycolic acid JRRR121, and glucose mycolate SMP70 gave an optimal specificity and sensitivity of 96/50 and 95/38, respectively, but trehalose dimycolates ZAM295 and ST123 both gave higher sensitivity/specificity combinations (94/75 and 94/63), respectively. The results for each serum sample with sets of antigens were combined using the R statistics package; this gave a specificity of 97.5% and sensitivity of 75% with two antigens (ST123 and JRRR121) and 100/69 if all six antigens were combined. 

### 3.2. Optimising the Distinction of All PCR-Positive Samples from PEI MAP Negatives

Given that the commercial serodiagnostic only gave a positive response with 16 of 80 PCR-positive samples, it was important to determine whether antibodies to mycolic acid-based lipids could be detected in more of the PCR-positive sera. The responses to the lipid antigens (Appendix A) were therefore re-analysed compared to the complete set of PCR positives. The optimal cut-offs for each antigen determined in this way and the optimal antigen combinations derived in R to provide the best distinction between the complete set of 80 PEI PCR-positive sera (PEI++ and PEI+− samples) and the 80 F5 and F6 ‘no-history’ negatives are given in Table 2. This shows the sensitivity for the 80 PCR-positive animals for each antigen, the specificity in each case being set to optimise the accuracy of match to PCR. The final two rows show the number of serum samples with each antigen or antigen combination at a fixed specificity of 97.5%. In this case, sensitivity is the percentage of all 80 PCR positive samples predicted to be positive in the lipid assay, and specificity is the percentage of no-history negatives identified as negative in the lipid assay (thus, for reference, the commercial IDEXX assay has a value of 20/100). The results with individual antigens were then combined in R so as to optimise the distinction between 80 PCR positives and 80 PCR-negative F5 and F6 samples. 

Thus, only two of the individual antigens identified more PCR-positive samples than IDEXX MAP ELISA: ZAM295, picking up two more but with a specificity reduced to 95%, and ST123, picking up five more but with a specificity reduced to 91%. However, the combination of the results with ST123 and JRRR121 identified 30 of the 80 PCR positive samples as positive in the assay, compared to just 16 using the IDEXX ELISA (the 16 PEI++ samples); in this case, the specificity was only 97.5%, compared to 100% for IDEXX ELISA. Of the 30 samples identified as positive in this way, 11 had PCR counts of 25.5–33 (n = 13), 7 from 33 to 37 (n = 23), 5 from 37 to 38.5 (n = 24), and 7 from 38.5 to 40 (n = 18).

An analysis of the fit of the above prediction of the lipid ELISA optimised using all 80 PCR positives to each sub-group of serum samples is presented in Appendix A. 

### 3.3. Determining Possible Effect of SICCT and M. bovis Infection and Vaccination on Lipid ELISA Performance

The optimal cut-offs for each antigen, as determined in Table 1, and the optimal antigen combinations derived in R (giving the results shown in Table 1) were applied to predict the positive or negative status of the additional sets of samples in Section 2.1 (ii, iii a–e, and iv); the results are presented in Table 3.

The distributions of the responses for each cohort of samples, including the F5 and F6 negatives and the PEI++ samples using single antigens ZAM295 and ST123 and two combinations of antigens, obtained using R statistics are shown in Figure 2. The corresponding distributions for other antigens and antigen combinations are shown in Appendix A.

Compared to the PEI++ MAP-positive samples, the MRI-negative sera gave a very low median, and all were predicted to be negative with all antigens except RT237F2 and all selected antigen combinations. 

For the SICCT-negative animals (APHA control 1–20), the lipid ELISA showed 100% specificity for all of the antigens except for RT237F2, and for all selected combinations of the antigens in R statistics. Similar results were obtained for the bTB reactors, except that one animal in twenty was false-positive with antigens ST123 and MOD171, and two with the ST123/JRRR121 combination.

Of the twenty Gudair-vaccinated cattle, one was detected as a false positive with antigen ST123, six with MOD171, and four with RT237F2. In this case, R values for specificity with the selected antigen combinations were 90–100%. 

For the BCG-vaccinated cattle, the lipid ELISA showed 100% specificity for three antigens (JRRR121, ZAM295, and ST123) and all selected antigen combinations in R. However, the BCG-vaccinated animals experimentally infected with *M. bovis* showed a somewhat reduced specificity (75–90%), although with antigen JRRR121, the specificity for MAP was 100%. Three antigen combinations using R statistics also showed 2–3 false positives. It is worth noting that ZAM295 on its own gave 100% specificity with all these sample sets except that from the vaccinated infected animals. 

With the serial serum samples from 10 MAP- and *M. bovis*-free animals taken over nine time-points (‘longitudinal’ samples), the lipid ELISA overall gave a specificity of 97% at the cut-off for PEI++ samples giving 62.5% sensitivity using antigen ST123, 91% specificity and 75% sensitivity with ZAM295, and 98% with the six-antigen combination in R using the cut-off to give 69% sensitivity. The individual responses are shown for each animal in Figure 3 for antigen ST123. Although very few of the responses are actually above the cut-off for a positive prediction, three of the animals do show persistent high responses. In two cases, AP2169 and AP2176, these animals also show somewhat elevated Bovigam PPDA responses (around 4 and 6 times, respectively, the averages for the remaining eight animals) (Appendix A); whether there is any link is not clear. 

## 4. Discussion

The diagnosis of Johne’s disease in cattle, particularly at an early stage, has proven challenging, not least because of the very slow development of the symptoms of clinical infection. Thus, the use of faecal PCR, or indeed faecal culture, is complicated by intermittent shedding, by sampling issues, by direct pass-through, and, arguably, by animals exposed but apparently able to resolve the initial infection. Indeed, a cut-off <33 PCR cycles of quantification for a confirmed infection has recently been suggested [17]. Serology has conventionally used quite high cut-offs in ELISA, leading to high specificity, but often to a rather low sensitivity. As seen from the 80 faecal PCR MAP-positive animals in this study, only 16 of which were IDEXX MAP-positive, there can be very poor agreement between these methods as to which animals are positive, though that agreement is better when PCR quantification cycles are low. In this case, the average cycle count for PEI++ samples (PCR-positive, MAP IDEXX-positive) was 27.6 (13 of 16 below 33), whereas for PEI+− samples (PCR-positive, MAP IDEXX-negative), it was 37.6 (none below 33). The use of histopathology at slaughter in following disease development and herd management is clearly limited. Analysis of results is further complicated by the different approaches to using these methods from country to country and between research groups, by herd infection and disease progression levels, and by inter-animal and husbandry variations. 

In this study, we set up an ELISA utilising six individual synthetic lipid antigens for the serodiagnosis of MAP infection. In the initial analysis comparing the serological responses of 16 MAP-infected animals positive both by PCR and IDEXX ELISA (PEI++) with 80 test-negative animals from two farms (F5 and F6), individual antigens ZAM295 and ST123 gave the best combination of specificity and sensitivity, with specificities and sensitivities of 93.8/75 and 93.8/62.5, respectively. Interestingly, combining the individual assay results using R statistics for these two antigens resulted in a reduced sensitivity of 56.3% at the same specificity of 93.8%; this may reflect the particular combination of false-positive responses, as the optimal cut-off was higher than with other combinations. However, the combination of the results for ST123 and JRRR121 gave the best specificity at 97.5% and sensitivity of 75%. Combining the results from all six individual antigens gave a higher specificity of 100% but lower sensitivity of 68.8%. It is possible that similar improvements might be achieved by mixing two or more of the antigens in a single ELISA; however, keeping the assays separate and then combining the results in a multiplex approach allows selective changes, such as different dominant antibody responses in different stages of infection, and indeed different infections to be recognised. Antigens MOD171 and RT237F2 generally gave low sensitivity and were less discriminative between MAP-infected animals, bTB reactors, Gudair- and BCG-vaccinated animals, and BCG-vaccinated and *M. bovis*-infected animals. 

The results show that there are some differences in response between the test-negative animals on the two PEI farms with no history of Johne’s disease. The animals from farm 5 were primarily housed outside, whereas those from farm 6 were housed inside exclusively, and therefore in closer proximity to each other and with different environmental exposures. Although farm 6 was a total confinement herd with no outdoor exposure, this farm did have a history of a large number of resident birds in the rafters of the barn and had a lower level of overall hygiene with increased stocking density. This particular barn had not been deep-cleaned for several years and there was significant detritus in the rafters that would occasionally drop down into the cattle living area, including the feed bunks and water troughs. It is therefore proposed that there may have been some cross-reactivity of the ELISAs with other environmental mycobacteria species that affected the observed specificity of the lipid ELISA in this herd.

Trehalose dimycolate ST123 gave the highest combination of sensitivity and specificity with the samples from farm 5; this antigen is a wax ester mycolic acid characteristic of MAC and not present in organisms such as *M. bovis* [35]. This antigen performed well with the samples from farm 6, though in this case, the best antigen was trehalose dimycolate ZAM295, more typical of faster-growing environmental mycobacteria. Antigen JRRR121 gave high specificity with both sets of MAP negative samples, albeit with only modest sensitivity for the PEI++ samples; this antigen is a *trans*-cyclopropane containing mycolic acid, again a sub-structure common in MAC [38]. The other three antigens showed rather lower sensitivities for the PEI++ samples at cut-offs giving as low as 91% specificity for F5 and F6 negatives. Antigen MOD171 is a model fragment from a mycobacterial cell wall, unlike the other non-wall-bound antigens used in this study, and may be less antigenic for that reason. The glucose ester SMP70 incorporates a methoxymycolic acid that is common in many mycobacterial cells and modest performance in the assay may reflect cross-reactivity with antibodies raised to other mycobacterial exposure/infection; it is worth noting that this class of mycolic acid is generally reduced in BCG vaccines [22]. Although antigen RT237F2 does contain a widely distributed mycolic acid, it also includes an amino-sugar and is not a known natural product; nonetheless, it was included in this study because it did produce very strong responses with some PCR-positive samples. Antigens SMP70, ST123, and ZAM295 are all examples of different classes of non-wall-bound trehalose mycolates, which are known to be strongly antigenic [22,23,25,26,27,28]. JRRR121 is a common class of free mycolic acid; these molecules are known to be antigenic, though responses are generally weaker than with trehalose or glucose esters [22,23,28].

It has been reported previously that the correlation between faecal PCR and serological assays may be poor at individual animal [39] or herd level [40], particularly in disease that has not reached a late stage. There have been similar studies on the agreement between faecal culture and serology [12,13]. With the current 80 PEI PCR-positive serum samples, there is a clear correlation between high IDEXX responses and low PCR cycle numbers—twelve of the sixteen samples that had positive IDEXX responses also had quantification cycles below 31, and none of those with IDEXX responses below the cut-off had quantification cycles below 33. There could be a number of reasons for the PCR-positive and IDEXX-negative samples, including direct pass-through of the organism. It was of interest, therefore, to see whether the lipid assay would provide a better, or different, fit than IDEXX to the complete set of PCR-positive data; in other words, whether the lipid assay gives more responses above a particular cut-off for the complete PCR-positive set than the 16 from IDEXX ELISA. In this study, two of the individual lipid antigens did identify more PCR-positive samples than the commercial ELISA: ZAM295 detected two more but with a specificity reduced to 95%, and ST123 detected five more but with a specificity reduced to 91%. However, the combination of the results with ST123 and JRRR121 identified 30 of the PCR positive samples as positive in the lipid ELISA assay, compared to just 16 using the IDEXX ELISA (the 16 PEI++ samples used in the initial analysis). In this case, the specificity was only 97.5% compared to 100% for the IDEXX ELISA; however, if the cut-off in the IDEXX ELISA was reduced to give 97.5% specificity, it only picked up 17 positives. The lipid ELISA could therefore be a good alternative to the commercial IDEXX ELISA for routine screening of cattle, particularly in the early stages of disease progression [41], and this would be worth investigating further. In the commercial PCR used in this study, samples were considered MAP-positive within 42 cycles. Of the 80 PCR-positive samples utilised in the analyses, 13 gave PCR quantification cycles <33, 38 between 33 and 38, 27 between 38 and 40, and 2 gave >40. A recent publication recommends that the maximum number of quantification cycles that can safely be interpreted as ‘infection’ is <33 and that a PCR cycle count of >33 and <40 should be regarded as inconclusive and requires a second confirmatory test [17]. Of the 30 samples identified as positive using the combination of the results for ST123 and JRRR121 in the R statistics package, only 11 had PCR counts of 25.5–33, and the others were spread across all quantification cycles from 33 to 40. Thus, the lipid ELISA identifies antibody responses in serum that appear to support infection or developing disease in a number of the high-cycle-count PCR positives.

All antigens except for RT237F2 correctly identified the APHA control animals as negative for MAP infection. Since these animals had been tested ten days previously with the statutory bTB skin test, this suggests that responses to these lipid antigens are unaffected by the skin test. The number of animals tested in this study is small, so further investigations are necessary to confirm this. Currently, cattle are not tested using a commercial serum ELISA within three months of a bTB skin test to avoid any cross-reactivity.

Responses to individual antigens JRRR121, SMP70, and ZAM295 and either a combination of all the antigens or ST123/ZAM295 were unaffected by Gudair vaccination eight weeks post vaccination. This suggests that the lipid ELISA has potential as a DIVA diagnostic. However, individual antigens MOD171, RT237, and ST123 and combinations ST123/JRRR121, ST123/JRRR121/ZAM295, and JRRR121/SMP70/ZAM295 were less specific, detecting a few false-positive animals. This may reflect changes in immune responses caused by vaccination selectively activating antibodies to other non-tuberculous mycobacteria that are recognised by these antigens, or could represent antibodies against MAP directly generated by vaccination with Gudair vaccine (inactivated strain of *M. paratuberculosis*) [22].

Responses to individual antigens ST123, ZAM295, and JRRR121 and to all selected antigen combinations were unaffected by vaccination with BCG. BCG vaccination is being investigated as a potential vaccine for bTB [41], and if found to be effective, it would likely be used extensively in countries with endemic bTB. It is therefore important to develop new MAP diagnostics with this in mind.

JRRR121 was found to have 100% specificity for animals infected with *M. bovis* (bTB reactors and BCG-vaccinated/*M. bovis*-infected animals). Other lipid antigens and selected antigen combinations gave reduced specificity: 90–95% in bTB reactors and 75–90% in BCG-vaccinated/*M. bovis*-infected animals. There were differences in the responses between these two groups, for which there could be several reasons. The bTB reactors represent natural *M. bovis* infections contracted in the field, whereas the animals in the BCG-vaccinated/*M. bovis*-infected group were infected experimentally with high doses of *M. bovis*. The animals in the bTB reactor group will be at different stages of infection to the experimentally infected animals. Additionally, in the BCG-vaccinated/*M. bovis*-infected group, the vaccination may have changed the animals’ immune response to infection, which may have resulted in a change in the antibody responses to the lipid antigens.

The lipid ELISA was used to evaluate the responses to the antigens over the course of a year in 10 animals, all of which remained negative for MAP infection, as determined by the IDEXX ELISA, and negative for *M. bovis*, as indicated by low IGRA responses. The specificity of the lipid ELISA was somewhat reduced. Using the combination of ST123/JRRR121, 93% specificity corresponded to six false positives; however, four of these were from a single animal at several time-points, possibly reflecting an early infection (by MAP or other mycobacteria) not detected by the IDEXX ELISA. This requires further investigation. 

These results suggest that assays detecting the binding of antibodies generated in MAP disease can be detected in ELISA with specific lipid antigens, or combinations of those assays, in more PCR-positive samples than when using a current commercial serodiagnostic device. Moreover, the assay is not interfered with by vaccination for MAP or bovine TB. On the basis that not all PCR positives are caused by direct pass-through, and some represent real disease, an assay that can detect which ones correspond to developing disease would be ideal for a diagnostic test. These initial results appear to support that possibility, but clearly require validation with much larger sample sets and with longitudinal sets of serum from animals known to be exposed to and infected with MAP.

## 5. Conclusions

This study provides evidence that ELISA using synthetic lipid antigens identical to individual components of mycobacterial cells can identify cattle that are positive for MAP infection by a combination of PCR and commercial MAP ELISA. The best antigens from the six evaluated here were a trehalose ester of a wax ester mycolic acid (ST123) characteristic of *Mycobacterium avium* complex, a free mycolic acid containing a *trans*-cyclopropane group common in *Mycobacteium avium* complex (JRRR121), and a trehalose ester of a diene containing mycolic acid class common in non-tuberculous mycobacteria (ZAM295). The assay is not interfered with by BCG vaccination for bTB, Gudair vaccination for MAP, or by a bTB skin test, and is not compromised by active bTB. Thus, it may find applications as a DIVA assay. It also provides a better correlation between PCR positivity and serodiagnosis than the commercial MAP IDEXX ELISA and may provide a useful alternative for routine screening of cattle for Johne’s disease, particularly in the early stages of disease.

## Figures and Tables

**Figure 1 animals-14-00848-f001:**
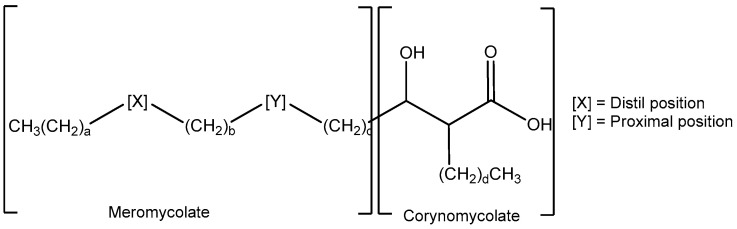
The generic structure of mycolic acids (MAs).

**Figure 2 animals-14-00848-f002:**
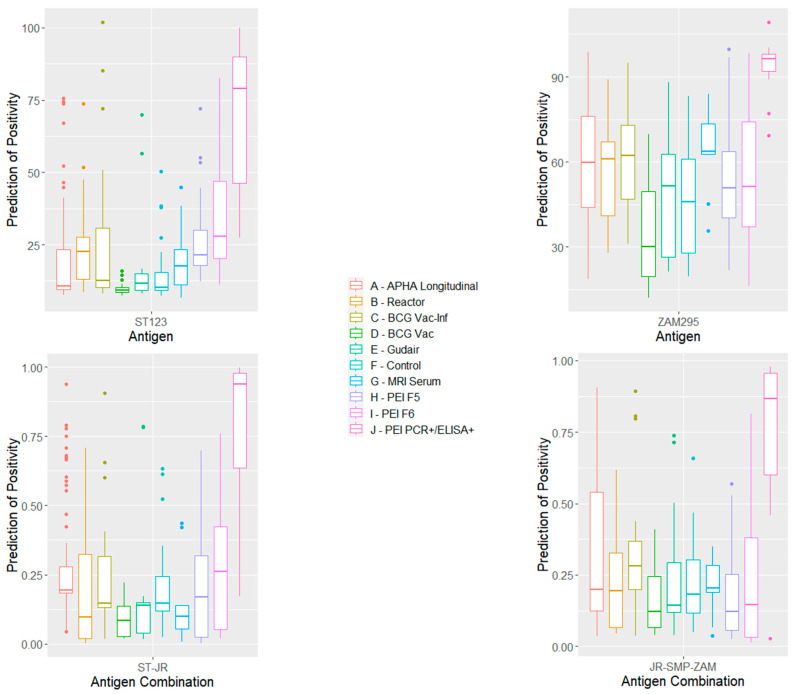
Box-and-whisker plots of the distribution of antigen responses with each cohort of MAP-negative samples and PEI++ samples (positive to PCR and IDEXX) to individual antigens ST123 and ZAM295, together with those to combinations of antigens ST123/JRRR121 and JRRR121/SMP70/ ZAM295 using the R statistics package. A–J refer to columns from left to right.

**Figure 3 animals-14-00848-f003:**
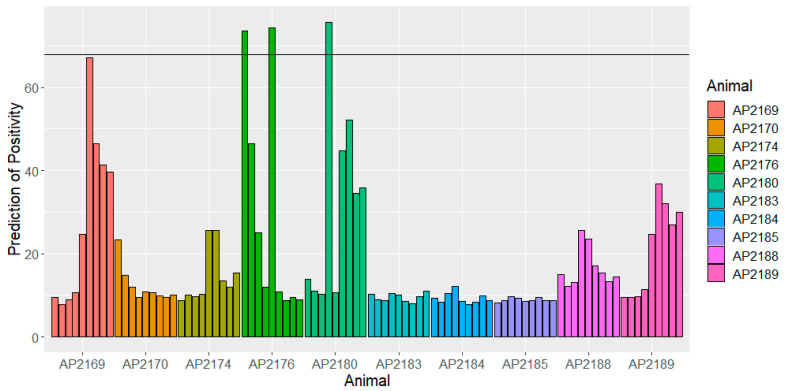
Responses for 10 Bovigam- and MAP IDEXX-negative animals in ELISA with antigen ST123, over 47 weeks (9 samples each) showing optimal cut-off for distinguishing PEI++ samples from 80 F5 and F6 negatives. The horizontal line corresponds to the cut-off from Table 3 for a positive diagnosis with this antigen.

**Table 1 animals-14-00848-t001:** Median responses for each set of samples with each of six antigens as % positive control for that antigen.

	Antigen or Antigen Combination Using R		MOD 171	JRRR 121	RT237 F2	SMP 70	ZAM 295	ST 123	ST123 + ZAM295	ST123 + JRRR121	ST123 + JRRR121 + ZAM295	JRRR121 + SMP70 + ZAM295	All Six Antigens
			% Pooled Pos						Probability of positive (0 to 1) with antigen combinations				
1	PEI++ n = 16 median		27.3	38.8	56.5	88.4	96.3	79.0	0.9	0.9	0.9	0.9	0.8
2	F5 plus F6 negative, n = 80 median		24.7	17.4	48.8	54.2	50.7	23.7	0.1	0.1	0.1	0.1	0.1
3	*p* * (F5 plus F6 compared to PEI++)		ns *	<0.0001	ns	<0.0001	<0.0001	<0.0001	<0.0001	<0.0001	<0.0001	<0.0001	<0.0001
			**Sensitivity analysis for single antigens**						**Sensitivity/specificity analysis for antigens** **combined in R**				
4	**Optimal cut-off for >93% specificity**		57	40	76	93	91	68	0.8	0.63	0.73	0.63	0.78
5	Optimal sensitivity %		25.0	50.0	6.3	37.5	75.0	62.5	56.3	75.0	68.8	68.8	68.8
6	Optimal specificity %		95.0	96.3	93.8	95.0	93.8	93.8	93.8	97.5	97.5	97.5	100.0
7	Accuracy % (sens + spec)/2		60.0	73.1	50.0	66.3	84.4	78.1	75.0	86.3	83.1	83.1	84.4
**Data for PEI++ relative to just F5 or F6 at cut-offs for above sensitivity**													
8	F5	Specificity %	97.5	92.5	92.5	95	95	97.5	97.5	100	97.5	100	100
9		*p* (F5 compared to PEI++)	<0.05	<0.0001	ns	<0.0001	<0.0001	<0.0001	<0.0001	<0.0001	<0.0001	<0.0001	<0.0001
10	F6	Specificity %	92.5	100	95	95	92.5	90	90	95	97.5	95	100
11		*p* (F6 compared to PEI++)	ns	<0.0001	ns	<0.01	<0.0001	<0.0001	<0.0001	<0.0001	<0.0001	<0.0001	<0.0001

Median responses for each set of samples with each of six antigens as % positive control for that antigen. Optimal cut-offs are based on 16 PEI PCR- and IDEXX-positive samples (PEI++) compared to all 80 PEI F5 and F6 negatives (again as % positive control), and sensitivity and specificity of each group are based on those cut-offs. In some cases, such as for MOD171, the optimal cut-off by ROC analysis led to a very low specificity; cut-offs were then identified to give maximum accuracy (average of sensitivity and specificity) at >93% specificity. The data for F5 and F6 samples are also presented separately, using the same cut-offs. The final five columns show the median calculated negative or positive status (on 0 to 1 scale) of PEI++ compared to all 80 F5 and F6 negative samples, combining results with several specified antigens using R statistics; the calculated values for each serum sample have then been analysed by ROC and optimal cut-offs determined to provide the highest accuracy (average of sensitivity and specificity). * Significance values were calculated using the Tukey test; ns = not significant.

**Table 2 animals-14-00848-t002:** Optimisation of results from lipid assay to predict status all 80 PEI PCR-positive samples compared to 80 F5 and F6 negatives, with single antigens and combinations of antigens in R.

		Antigen						Antigen Combination					
		MOD 171	JRRR 121	RT 237 F2	SMP 70	ZAM 295	ST 123	ST123 + ZAM295	ST123 + JRRR121	SMP70 + JRRR121 + ZAM295	ST123 + JRRR121 + ZAM295	All Six Antigens	MAP IDEXX
		% Pooled Pos						Samples Meeting Cut-Off for Positive Diagnosis					
PCR+ n = 80	Sens %	10.0	20.0	8.8	12.5	22.5	26.3	22.5	37.5	38.8	38.8	36.3	20
Number matching PCR		8	16	7	10	18	21	18	30	31	31	29	16
PEI ‘no-history’ negatives													
F5 + F6 n = 80	Spec %	92.5	92.5	96.3	91.3	95.0	91.3	95.0	97.5	95.0	95.0	95.0	100
Compared to PCR+		ns	ns	ns	ns	ns	ns	ns	ns	ns	ns	ns	ns
F5 n = 40	Spec %	95	97.5	95	95	92.5	87.5	92.5	97.5	92.5	92.5	95	100
*p* (compared to PEI+)		ns	<0.05	ns	<0.0001	<0.001	<0.05	<0.001	<0.001	<0.0001	0.0001	<0.0001	0.05
F6 n = 40	Spec %	90	87.5	97.5	87.5	97.5	95	97.5	97.5	95	97.5	95	100
*p* (compared to PEI+)		ns	<0.05	<0.05	ns	<0.001	ns	<0.05	<0.05	<0.001	<0.05	<0.05	<0.05
Using a fixed specificity of 97.5%													
PCR+ sens.	Sens %	5	17.5	7.5	7.5	11.3	17.5	17.5	37.5	33.8	18.9	30	21.3
Number matching PCR PEI+		4	14	6	6	9	14	14	30	27	15	24	17

The optimal sensitivity and specificity, assuming the 80 PCR positive samples are true positives, for the two sets of ‘no-history’ negatives from farms F5 and F6, and the 80 combined F5 plus F6 PEI samples. Data for sets of antigens are derived by optimising the combinations of antigen responses to provide the best distinction between 80 PCR positive and PCR negative samples using the R statistics package. Significance values were calculated using the Tukey test; ns = not significant. The final column is the result with a commercial IDEXX assay; the number of predicted positives in IDEXX, in comparison to the lipid assay, at 97.5% specificity was calculated by reducing the cut-off to reach that specificity with the 80 F5 and F6 negatives using the data in Appendix A. An analysis of the data for other sample cohorts using these cut-offs and antigen combinations optimised for all 80 PCR positives is shown in Appendix A.

**Table 3 animals-14-00848-t003:** Median responses (as % positive control) and sensitivity/specificity for Moredun ‘no-history’ samples and for each cohort of APHA serum samples compared to 16 PEI PCR- and IDEXX-positive samples using the same cut-offs as determined in Table 1.

			Antigen						Antigen Combination Using R				
			MOD 171	JRRR 121	RT237 F2	SMP 70	ZAM 295	ST 123	ST123 + ZAM295	ST123 + JRRR121	ST123 + JRRR121 + ZAM295	JRRR121 + SMP70 + ZAM295	All Six Antigens
		Cut-off	57	40	76	93	91	68	0.8	0.63	0.73	0.63	0.78
PEI++ (as per Table 1)		Sens %	25.0	50.0	6.3	37.5	75.0	62.5	56.3	75.0	68.8	68.8	68.8
	n	Group in Section 2.1	Specificity based on single antigens						Specificity based on R combinations				
Moredun serum	9	(ii)	100.0	100.0	88.9	100.0	100.0	100.0	100.0	100.0	100.0	100.0	100.0
*p* (compared to PEI++)			ns	<0.05	ns	<0.001	<0.001	<0.001	0.0001	<0.001	0.0001	0.0001	<0.001
APHA controls	20	(iii) (a)	100.0	100.0	75.0	100.0	100.0	100.0	100.0	100.0	100.0	95.0	100.0
*p* (compared to PEI++)			ns	<0.01	ns	<0.001	<0.0001	<0.0001	<0.0001	<0.0001	<0.0001	<0.0001	<0.0001
bTB reactors	20	(iii) (b)	95.0	100.0	95.0	100.0	100.0	95.0	100.0	90.0	100.0	100.0	100.0
*p* (compared to PEI++)			ns	<0.001	<0.01	<0.001	<0.0001	<0.0001	<0.0001	<0.0001	<0.0001	<0.0001	<0.0001
Gudair-vaccinated	20	(iii) (c)	70.0	100.0	80.0	100.0	100.0	95.0	100.0	90.0	95.0	90.0	100.0
*p* (compared to PEI++)			ns	<0.01	ns	<0.001	<0.0001	<0.0001	<0.0001	<0.0001	<0.0001	<0.0001	<0.0001
BCG-vaccinated	20	(iii) (d)	95.0	100.0	85.0	95.0	100.0	100.0	100.0	100.0	100.0	100.0	100.0
*p* (compared to PEI++)			ns	<0.001	ns	<0.001	<0.0001	<0.0001	<0.0001	<0.0001	<0.0001	<0.0001	<0.0001
BCG vac-inf	20	(iii) (e)	90.0	100.0	85.0	75.0	90.0	85.0	90.0	90.0	85.0	85.0	90.0
*p* (compared to PEI++)			ns	<0.01	ns	<0.05	<0.0001	<0.001	<0.0001	<0.0001	<0.0001	<0.0001	<0.0001
Longitudinal samples: 10 ParaTB IDEXX-negative animals, each over 9 times (total 47 weeks)	90	(iv)	97.8	98.9	93.3	96.7	91.1	96.7	94.4	90.0	93.3	85.6	97.8
*p* (compared to PEI++)			<0.05	<0.0001	ns	<0.0001	<0.0001	<0.0001	<0.0001	<0.0001	<0.0001	<0.0001	<0.0001

Median responses (as % positive control) and sensitivity/specificity for Moredun ‘no-history’ samples and for each cohort of APHA serum samples (Section 2.1), using the same cut-offs as determined in Table 1. PEI++ data show the corresponding sensitivity with the 16 PEI samples both positive to PCR and giving IDEXX responses above the recommended cut-off for MAP infection. Specificity values for single antigens are determined from Appendix A using the cut-offs from Table 1. Specificity values for combinations of antigens using R for each cohort are calculated using the antigen combinations determined in Table 1; the cut-offs for R calculations are as in Table 1 and are on a scale 0 (negative) to 1 (positive). Significance values were calculated using the Tukey test; ns = not significant. Individual cohorts are Moredun ‘no-history’ MAP negatives, APHA controls, BTB reactor cattle, Gudair-vaccinated cattle, BCG-vaccinated cattle, and BCG-vaccinated but M. bovis-infected cattle. The final set of 90 ‘longitudinal’ serum samples are from 10 control animals, each at 9 times over 47 weeks, and negative by MAP IDEXX at each time.

## Data Availability

All data are provided as part of the Appendix A.

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
