# Peer review of "An ELISA Using Synthetic Mycolic Acid-Based Antigens with DIVA Potential for Diagnosing Johne’s Disease in Cattle"

_animals, 2024, doi:10.3390/ani14060848_

Round 1

Reviewer 1 Report

Comments and Suggestions for Authors

An ELISA using synthetic mycolic acid-based antigens with DIVA potential for diagnosis Johne’s disease.

This study reports the development of a new serologic test based on different antibodies' detection than the previous serologic tests. The recommendation of OIE to develop a new test was followed by assessing performance in positive control (case), negative control (control), and potential factors of variation in performance. The results are promising for accomplishing the next steps of development and testing accuracy, precision, robustness, and then, performance in different populations. I think that this new diagnostic test is promising, however, I had major comments before recommending publication.

Major comments:

1-     This study is the first step development of a diagnostic test. This characteristic is not mentioned in the manuscript. I recommend the authors take knowledge of this very interesting paper from OIE about steps to follow to develop a new diagnostic test. I recommend adding a paragraph in the intro to describe briefly these different steps and to clarify to the readers that it is the first step and results should be taken with cautious before further development.

Diaz F. 2014. OIE Standard on principles and methods of validation of diagnostic assays

for infectious diseases. In: Proceedings of the OIE Regional Workshop for OIE National Focal Points for Veterinary Products, Tokyo, Japan 2014 Dec 3 (pp. 3-5).

2-     I found that the plus-value to develop this test in comparison with existing tests is not put enough forward. Indeed, the authors talk about limits of the PCR but they do not explain why a more performance serologic test should be developed. What the current IDEXX ELISA are measuring? What is its current performance? Why the change in antibody targeting’s going to improve the diagnosis?

3-     Finally, the authors compared different Se and Sp but there is no information about precision around the finding values (95% confidence intervals). It would be easier to know the distribution to assess if one or another Ag is really better or not.

Minor comments:

1-     Line 3 (Title): It is not obvious to every reader what DIVA means. I recommend rewording the title including the type of study (development of a new diagnostic test) and removing the acronym DIVA.  

2-     Line 22: The simple summary is supposed to be a short summary of the abstract. However, the simple summary is longer than the abstract in this study. I recommend shortening this paragraph.

3-     Line 41: what do the authors mean by "agreement"? between tests? In positive and or negative cases? inter or intra-operator? Please, clarify.

4-     Line 41: “The potential solution”. Go at the line.

5-     Line 44: Define DIVA.

6-     Line 44: Methods – information about statistical analysis, and the number of sera in the positive group and negative group is missing.

7-     Line 77: Uniformize unity of measurement (days or months).

8-     Line 81: What do the authors mean by pathology? macroscopic assessment, histologic? Please, clarify.

9-     Line 84: Do the authors have any reference to justify this statement?

10- Line 90Is it really indirect? PCR detects the agent directly, is it not the response of the organism?

11- Line 101-102: Please define what is the sensitivity and specificity of a diagnostic test.

12- Line 103: Why the sentence “on its own” is it in bold?

13- Line 122: The authors use M.Bovis for Mycobacterium Bovis throughout the manuscript. It could be confusing with Mycoplasma Bovis. I suggest writing the complete name in the text.

14- Line 178: I suggest adding a separate paragraph for “potential factors of variations”.

15- Line 234: Additional information about IDEXX ELISA is necessary to judge why this new test could be more advantageous (which Ag? Se? Sp? Any factors of variation? etc.)

16- Line 236: Has this test been validated in cattle or other species?

17- Line 269: I recommend adding justification about why doing separate validation in positive PCR sera.

18- Line 310-311: “However, …study”. I’m not sure to understand why those antigens were kept. Could you rewrite this sentence to clarify its meaning?

19- Line 322: Why 91% and not 93%?

20- Line 327: Is it Table 2 or 3?

21- Line 367 (Figure 2): What do the authors mean by the prediction of positivity? Se? positive predictive value? Please, clarify.

22- Line 412: As I mentioned in the intro, what do the authors mean by “agreement”?

23- Line 425: What do the authors mean by “using R statistic”? Doing combination?

24- Line 472-473: I do not think this is true for all statuses of the disease. The performance of both tests is similar when an animal has obvious clinical signs.

25- Line 495-496: “only 11… (n=13). Is it 11 or 13?

26- Line 498-499: This sentence is more a result that a discussion.

27- Line 532-537: Is it any reference or it is just a part of this study?

28- Line 545: The step to follow after this step of development needs to be mentioned as the next step.

Author Response

Referee 1

An ELISA using synthetic mycolic acid-based antigens with DIVA potential for diagnosis Johne’s disease.

  1. This study reports the development of a new serologic test based on different antibodies' detection than the previous serologic tests. The recommendation of OIE to develop a new test was followed by assessing performance in positive control (case), negative control (control), and potential factors of variation in performance. The results are promising for accomplishing the next steps of development and testing accuracy, precision, robustness, and then, performance in different populations. I think that this new diagnostic test is promising, however, I had major comments before recommending publication.

We are pleased that the referee thinks the work is promising.

Major comments:

  1. This study is the first step development of a diagnostic test. This characteristic is not mentioned in the manuscript. I recommend the authors take knowledge of this very interesting paper from OIE about steps to follow to develop a new diagnostic test. I recommend adding a paragraph in the intro to describe briefly these different steps and to clarify to the readers that it is the first step and results should be taken with cautious before further development.

Diaz F. 2014. OIE Standard on principles and methods of validation of diagnostic assays

for infectious diseases. In: Proceedings of the OIE Regional Workshop for OIE National Focal Points for Veterinary Products, Tokyo, Japan 2014 Dec 3 (pp. 3-5).  The reference has been included.

  1. I found that the plus-value to develop this test in comparison with existing tests is not put enough forward. Indeed, the authors talk about limits of the PCR but they do not explain why a more performance serologic test should be developed. What the current IDEXX ELISA are measuring? What is its current performance? Why the change in antibody targeting’s going to improve the diagnosis? Text added at 118-121, 155-159

  1. Finally, the authors compared different Se and Sp but there is no information about precision around the finding values (95% confidence intervals). It would be easier to know the distribution to assess if one or another Ag is really better or not. We have added significance values to Tables 1 and 3 that show the differences in performance between the antigens. The RoC curves presented in the supplementary include confidence levels. We hope this meets the requirements of the referee. If the referee is asking about reproducibility from run to run using the same samples, and between different operators, then we have not carried out such studies on the sample sets described in the paper – the results we present are at the proof of concept stage and not those of a fully developed commercial assay (rather an initial stage as per the comment 2 above.

Minor comments:                                                             

  1. Line 3 (Title): It is not obvious to every reader what DIVA means. I recommend rewording the title including the type of study (development of a new diagnostic test) and removing the acronym DIVA.  We would prefer not to do this as there are many published papers that use the acronym, and to use the full words makes the title rather unwieldy.
  2. Line 22: The simple summary is supposed to be a short summary of the abstract. However, the simple summary is longer than the abstract in this study. I recommend shortening this paragraph. That is not how we read the instructions for authors – the simple summary is targeted to a lay audience. Both are somewhat longer than the 200 word advice, but the simple summary is actually shorter. We would prefer to leave these as they stand unless there is an editorial requirement to shorten them.
  3. Line 41: what do the authors mean by "agreement"? between tests? In positive and or negative cases? inter or intra-operator? Please, clarify. Faecal PCR or culture give far more positive responses than IDEXX ELISA. and agreement as to which sample is positive is only good when cycle counts are low and IDEXX values are high (the recommended cut-off in ELISA is actually set rather high). By that time, arguably some animals will be showing visible signs of disease. Faecal PCR and culture do give responses in what might be thought to be early stage infection, but there is no certainty that these reflect true infection because there is no quantification of the number of direct pass-through cases. One major aim of this work is to see whether lipid antigens can support some of these PCR/culture results as representing true infection or disease.
  4. Line 41: “The potential solution”. Go at the line. Sorry, I don’t understand what this means.
  5. Line 44: Define DIVA. Done
  6.   Line 44: Methods – information about statistical analysis, and the number of sera in the positive group and negative group is missing.

Line 44 strictly refers to the methods in the abstract. This is already longer than recommended – the numbers of animals can be added but we would prefer not to make it even longer unless required by the editorial staff.

 Additional statistical data have been added to the methods section of the paper and the methods used have been added to the methods section. Numbers of sera in each group are now included in the methods.

  1. Line 77: Uniformize unity of measurement (days or months). Done

12.Line 81: What do the authors mean by pathology? macroscopic assessment, histologic? Please, clarify. Part of this text are the definitions from the reference. However, a sentence has been added.

  1. Line 84: Do the authors have any reference to justify this statement? Reference added.
  2. Line 90Is it really indirect? PCR detects the agent directly, is it not the response of the organism? Indirect has been removed – people may treat it as direct evidence of infection, but as the following sentence states, it is only direct evidence of exposure as (unspecified) numbers of direct pass through may occur.
  3. Line 101-102: Please define what is the sensitivity and specificity of a diagnostic test. We will do this is required, but the definitions are so well known that we think it is not necessary for this special issue.
  4. Line 103 (97?) Why the sentence “on its own” is it in bold? The bold has been removed
  5. Line 122: The authors use M.Bovis for Mycobacterium Bovis throughout the manuscript. It could be confusing with Mycoplasma Bovis. I suggest writing the complete name in the text. We have explicitly defined the term instead – we trust this is acceptable.
  6. Line 178: I suggest adding a separate paragraph for “potential factors of variations”. We do not really understand this comment at this line, or close to it.
  7. Line 234: Additional information about IDEXX ELISA is necessary to judge why this new test could be more advantageous (which Ag? Se? Sp? Any factors of variation? etc.) Additional text added.
  8. Line 236: Has this test been validated in cattle or other species? Additional description provided.
  9. Line 269: I recommend adding justification about why doing separate validation in positivee PCR sera. This has been added at lines 480-490 – it seems more appropriate in the discussion than in the methods.
  10. Line 310-311: “However, …study”. I’m not sure to understand why those antigens were kept. Could you rewrite this sentence to clarify its meaning? The sentence has been changed.
  11. Line 322: Why 91% and not 93%? Removed
  12. Line 327: Is it Table 2 or 3? Changed in line 330.
  13. Line 367 (Figure 2): What do the authors mean by the prediction of positivity? Se? positive predictive value? Please, clarify. Not sure about the line number and there is no Figure 2, but the heading of Table 2 has been edited.
  14. Line 412: As I mentioned in the intro, what do the authors mean by “agreement”? Some additional words have been added which hopefully clarify this.
  15. Line 425: What do the authors mean by “using R statistic”? Doing combination? I don’t understand this as the words before say just that.
  16. Line 472-473: I do not think this is true for all statuses of the disease. The performance of both tests is similar when an animal has obvious clinical signs. Actually lines 482-483 -this has been adjusted and an additional reference included. There is published work that shows only 70 pc agreement between faecal culture and serology even in late stage disease.
  17. Line 495-496: “only 11… (n=13). Is it 11 or 13? Corrected.
  18. Line 498-499: This sentence is more a result that a discussion. An additional sentence has been added.
  19. Line 532-537: Is it any reference or it is just a part of this study? The results are part of this work, but a reference has been added.
  20. Line 545: The step to follow after this step of development needs to be mentioned as the next step. Text added.

Reviewer 2 Report

Comments and Suggestions for Authors

The aim of the paper is to give a contribution to improve quality of indirect diagnosis of paratuberculosis in cattle, in particular to find a method more sensitive than a commercial kit and without interference from vaccination either for MAP or for bovine TB. The study compares results of an ELISA commercial kit with an ELISA using different classes of single synthetic mycolic acid based antigens; the results seem promising.

The paper is very complicated to follow. In order to ameliorate its readability, firstly I would suggest to remove the last part refereed to the begins of a longitudinal study that the authors, at rows 544-545 stated will be the subject of a further publication. Secondarily, since even the first part of the study it is not easy to follow, I would suggest more schematisation of the set of samples tested, perhaps with a tabular summary at the end of Chapter 2.1.

The paper sets the context of the study well by summarising the characteristics of the disease and the diagnostic problems, in particular the possibility of identifying animals in the early stages of infection. The topic is surely relevant and the paper well identify the gap in knowledge. Even if the references are complete and appropriate, there are a couple of missing papers relative to the PCR diagnostic test that in my opinion the authors should consider, these are:

Kralik P, Slana I, Kralova A, Babak V, Whitlock RH, Pavlik I. Development of a predictive model for detection of Mycobacterium avium subsp. paratuberculosis in faeces by quantitative real time PCR. Vet Microbiol. 2011 Apr 21;149(1-2):133-8. doi: 10.1016/j.vetmic.2010.10.009. Epub 2010 Oct 21. PMID:21075565.

Russo S, Cortimiglia C, Filippi A, Palladini G, Garbarino C, Massella E, Ricchi M. Validation of digital PCR assay for the quantification of Mycobacterium avium subsp. paratuberculosis in bovine faeces according to the ISO 20395:2019. J Microbiol Methods. 2023 Oct;213:106825. doi: 10.1016/j.mimet.2023.106825. Epub 2023 Sep 20. PMID: 37739126.

Specific comments 

Rows 37 and 60 and rows 532-545: because of my above comment and since the authors stated this will be the subject of a further communication, I would remove any parts relative to the timeline of detection of the developed assay. Moreover, the data herein showed are not so convincing, as already discussed by the authors at rows 532-545 and, because of the impact they can have on the diagnosis of paratuberculosis, probably they deserve much more data.

Rows 78-83: I suppose here the authors are referring to reference 7 about the staging of the disease. Anyway, please report it also at the end of row 83, bearing in mind that, there are also other papers that suggested other definition of the stages, although more or less they are reporting the same concepts.

Rows 82-84: I disagree with the authors about this statement: although necropsy and testing of the recovered materials is generally considered the final answer for detecting Johne's disease, clinical sigs coupled with highly specific assays, are now generally accepted. For sure, the word “Pathology” here sounds a little bit unappropriated.

Rows 90-91: do the authors could suggest a direct evidence of infection in this field? This sentence is a little bit confusing, even because in the framework of veterinary diagnosis, test aimed at detecting the presence of antigens of the causative agent are considered as direct test, while those aimed at detecting antibodies are defined as indirect test. Please reformulate.

Rows 101-102: This sentence is not so precise, I would remember to the authors how, according to all papers dedicated to the validation of Johne's disease diagnostic tools, the sensitivity is low, but strictly depending on the onset of the disease. Please reformulate.

Rows 103-107: I would be very careful about this sentence, even if I have understood that the authors wanted to discuss that often is difficult/impossible discriminating between very low excreting MAP subject to false positive calls by PCR, the cycle of count (better say cycle of quantification according to the MIQE guidelines) are not absolute values and are extremely depending on the run, even in assay very well validated. For this reason, in order to check if the run was acceptable, negative and positive controls (these last should also have a confidence interval of acceptability) are included in each run. I would suggest to reformulate these sentences keeping in mind the above-mentioned concepts.

Row 148: Serum Sample. Information about the sera used in the longitudinal study is not present in this section, according to my suggestion I would remove this last part from the manuscript.

Rows 173-174: Please, did the herd introduce animals from farms with lower Johne's disease status?

Rows 208: Please remove the space after 2.2.

Row 237 : Please report here where or from who the lipids have been produced/purchased.

Row 300:  Culture was not mentioned in this paper, please correct with PCR.

Table 1 and elsewhere: why values are reported as median and not as mean? This is a generally accepted rule in the field of serological test? Because as mean the differences resulted less evident? Too much variability despite all the efforts to reduce it? I’m referring to what reported at rows 260-264.

Table 1 is generally a little bit hard to follow, I suspect the authors have already tried different ways to show these data and this is the best they were able to obtain, so I don’t have so much suggestions to give to theme but the Table 1 still remains difficult to understand at first impact.

Row 339: Please add a space after “only”.

Row 352: Please check if the samples are positive to culture or PCR (as reported in row 156).

Rows 388-399: Please, for the above-mentioned reasons, remove this part.

Row 412: Please check if you are speaking about  culture or PCR.

Row 408: I would suggest to reformulate taking into account the point discussed before about the cycles of quantification.

Row 442 and after: I cannot understand why animals housed inside should have much different environmental exposures than those housed outside. Please reformulate considering also what has been stated after.

Row 471: Please add a “dot” at the end of the sentence.

Row 488: Please remove “early stages of infection” according to what already suggested above. For the same reason, remove from rows 532 to 545.

Row 516: Please remove a dot and space.

Row 561: Please add  “.

Row 571: Please standardize characters (half in bold half not).

The aim of the paper is to give a contribution to improve quality of indirect diagnosis of paratuberculosis in cattle, in particular to find a method more sensitive than a commercial kit and without interference from vaccination either for MAP or for bovine TB. The study compares results of an ELISA commercial kit with an ELISA using different classes of single synthetic mycolic acid based antigens; the results seem promising.

The paper is very complicated to follow. In order to ameliorate its readability, firstly I would suggest to remove the last part refereed to the begins of a longitudinal study that the authors, at rows 544-545 stated will be the subject of a further publication. Secondarily, since even the first part of the study it is not easy to follow, I would suggest more schematisation of the set of samples tested, perhaps with a tabular summary at the end of Chapter 2.1.

The paper sets the context of the study well by summarising the characteristics of the disease and the diagnostic problems, in particular the possibility of identifying animals in the early stages of infection. The topic is surely relevant and the paper well identify the gap in knowledge. Even if the references are complete and appropriate, there are a couple of missing papers relative to the PCR diagnostic test that in my opinion the authors should consider, these are:

Kralik P, Slana I, Kralova A, Babak V, Whitlock RH, Pavlik I. Development of a predictive model for detection of Mycobacterium avium subsp. paratuberculosis in faeces by quantitative real time PCR. Vet Microbiol. 2011 Apr 21;149(1-2):133-8. doi: 10.1016/j.vetmic.2010.10.009. Epub 2010 Oct 21. PMID:21075565.

Russo S, Cortimiglia C, Filippi A, Palladini G, Garbarino C, Massella E, Ricchi M. Validation of digital PCR assay for the quantification of Mycobacterium avium subsp. paratuberculosis in bovine faeces according to the ISO 20395:2019. J Microbiol Methods. 2023 Oct;213:106825. doi: 10.1016/j.mimet.2023.106825. Epub 2023 Sep 20. PMID: 37739126.

Specific comments 

Rows 37 and 60 and rows 532-545: because of my above comment and since the authors stated this will be the subject of a further communication, I would remove any parts relative to the timeline of detection of the developed assay. Moreover, the data herein showed are not so convincing, as already discussed by the authors at rows 532-545 and, because of the impact they can have on the diagnosis of paratuberculosis, probably they deserve much more data.

Rows 78-83: I suppose here the authors are referring to reference 7 about the staging of the disease. Anyway, please report it also at the end of row 83, bearing in mind that, there are also other papers that suggested other definition of the stages, although more or less they are reporting the same concepts.

Rows 82-84: I disagree with the authors about this statement: although necropsy and testing of the recovered materials is generally considered the final answer for detecting Johne's disease, clinical sigs coupled with highly specific assays, are now generally accepted. For sure, the word “Pathology” here sounds a little bit unappropriated.

Rows 90-91: do the authors could suggest a direct evidence of infection in this field? This sentence is a little bit confusing, even because in the framework of veterinary diagnosis, test aimed at detecting the presence of antigens of the causative agent are considered as direct test, while those aimed at detecting antibodies are defined as indirect test. Please reformulate.

Rows 101-102: This sentence is not so precise, I would remember to the authors how, according to all papers dedicated to the validation of Johne's disease diagnostic tools, the sensitivity is low, but strictly depending on the onset of the disease. Please reformulate.

Rows 103-107: I would be very careful about this sentence, even if I have understood that the authors wanted to discuss that often is difficult/impossible discriminating between very low excreting MAP subject to false positive calls by PCR, the cycle of count (better say cycle of quantification according to the MIQE guidelines) are not absolute values and are extremely depending on the run, even in assay very well validated. For this reason, in order to check if the run was acceptable, negative and positive controls (these last should also have a confidence interval of acceptability) are included in each run. I would suggest to reformulate these sentences keeping in mind the above-mentioned concepts.

Row 148: Serum Sample. Information about the sera used in the longitudinal study is not present in this section, according to my suggestion I would remove this last part from the manuscript.

Rows 173-174: Please, did the herd introduce animals from farms with lower Johne's disease status?

Rows 208: Please remove the space after 2.2.

Row 237 : Please report here where or from who the lipids have been produced/purchased.

Row 300:  Culture was not mentioned in this paper, please correct with PCR.

Table 1 and elsewhere: why values are reported as median and not as mean? This is a generally accepted rule in the field of serological test? Because as mean the differences resulted less evident? Too much variability despite all the efforts to reduce it? I’m referring to what reported at rows 260-264.

Table 1 is generally a little bit hard to follow, I suspect the authors have already tried different ways to show these data and this is the best they were able to obtain, so I don’t have so much suggestions to give to theme but the Table 1 still remains difficult to understand at first impact.

Row 339: Please add a space after “only”.

Row 352: Please check if the samples are positive to culture or PCR (as reported in row 156).

Rows 388-399: Please, for the above-mentioned reasons, remove this part.

Row 412: Please check if you are speaking about  culture or PCR.

Row 408: I would suggest to reformulate taking into account the point discussed before about the cycles of quantification.

Row 442 and after: I cannot understand why animals housed inside should have much different environmental exposures than those housed outside. Please reformulate considering also what has been stated after.

Row 471: Please add a “dot” at the end of the sentence.

Row 488: Please remove “early stages of infection” according to what already suggested above. For the same reason, remove from rows 532 to 545.

Row 516: Please remove a dot and space.

Row 561: Please add  “.

Row 571: Please standardize characters (half in bold half not).

Author Response

Referee 2

  1. The aim of the paper is to give a contribution to improve quality of indirect diagnosis of paratuberculosis in cattle, in particular to find a method more sensitive than a commercial kit and without interference from vaccination either for MAP or for bovine TB. The study compares results of an ELISA commercial kit with an ELISA using different classes of single synthetic mycolic acid based antigens; the results seem promising.

Thank you for these positive comments.

  1. The paper is very complicated to follow. In order to ameliorate its readability, firstly I would suggest to remove the last part refereed to the begins of a longitudinal study that the authors, at rows 544-545 stated will be the subject of a further publication. We accept that we have complicated matters by bringing in reference to timeline studies on animals from infected herds, and that this has confused the discussion of the 90 samples from negative animals. We have removed those references but, with the referee’s agreement, would prefer to retain the 90 negatives – the responses may not be a perfect match for IDEXX, but they are what we got, and we think it is more rigorous to include them. We have changed the text describing them significantly. Clearly in taking this assay further we would need to identify whether these elevated responses are real (ie. the animal is generating antibodies to MAP) or caused by something else (perhaps a cross reaction to other organisms containing mycolic acids.

Secondarily, since even the first part of the study it is not easy to follow, I would suggest more schematisation of the set of samples tested, perhaps with a tabular summary at the end of Chapter 2.1. This has been done with the introduction of sub-headings. We hope this is now clearer.

  1. The paper sets the context of the study well by summarising the characteristics of the disease and the diagnostic problems, in particular the possibility of identifying animals in the early stages of infection. The topic is surely relevant and the paper well identify the gap in knowledge. Even if the references are complete and appropriate, there are a couple of missing papers relative to the PCR diagnostic test that in my opinion the authors should consider, these are:

Kralik P, Slana I, Kralova A, Babak V, Whitlock RH, Pavlik I. Development of a predictive model for detection of Mycobacterium avium subsp. paratuberculosis in faeces by quantitative real time PCR. Vet Microbiol. 2011 Apr 21;149(1-2):133-8. doi: 10.1016/j.vetmic.2010.10.009. Epub 2010 Oct 21. PMID:21075565.

Russo S, Cortimiglia C, Filippi A, Palladini G, Garbarino C, Massella E, Ricchi M. Validation of digital PCR assay for the quantification of Mycobacterium avium subsp. paratuberculosis in bovine faeces according to the ISO 20395:2019. J Microbiol Methods. 2023 Oct;213:106825. doi: 10.1016/j.mimet.2023.106825. Epub 2023 Sep 20. PMID: 37739126.
Thank you for the suggestion – the references have been added.

Specific comments 

  1. Rows 37 and 60 and rows 532-545: because of my above comment and since the authors stated this will be the subject of a further communication, I would remove any parts relative to the timeline of detection of the developed assay. Moreover, the data herein showed are not so convincing, as already discussed by the authors at rows 532-545 and, because of the impact they can have on the diagnosis of paratuberculosis, probably they deserve much more data. We are not comfortable to do this, though we accept that the mention of timeline studies on animals from infected herds has made the text complicated. It is those latter data which will be the subject of further submissions, not the results from the 10 negative animals that are reported here. We have therefore modified the text to remove mention of the timeline studies on infected herds, and adjusted the discussion of the 90 IDEXX negative samples in this manuscript. We hope this will be acceptable to the referee.
  2. Rows 78-83: I suppose here the authors are referring to reference 7 about the staging of the disease. Anyway, please report it also at the end of row 83, bearing in mind that, there are also other papers that suggested other definition of the stages, although more or less they are reporting the same concepts. A statement has been added.
  3. Rows 82-84: I disagree with the authors about this statement: although necropsy and testing of the recovered materials is generally considered the final answer for detecting Johne's disease, clinical sigs coupled with highly specific assays, are now generally accepted. For sure, the word “Pathology” here sounds a little bit unappropriated. We have adjusted the text somewhat to reflect these comments. We have replaced ‘pathology’ by ‘gross pathology or histopathology’ as per reference 7.
  4. Rows 90-91: do the authors could suggest a direct evidence of infection in this field? This sentence is a little bit confusing, even because in the framework of veterinary diagnosis, test aimed at detecting the presence of antigens of the causative agent are considered as direct test, while those aimed at detecting antibodies are defined as indirect test. Please reformulate. We recognise that an antibody assay is less direct than identifying the organism; however, we would argue that the fact that direct pass through is reported means that PCR/culture is only a DIRECT measure of exposure, not of infection. In that sense, a serodiagnostic assay detecting antibodies in serum could actually be seen to be a more direct measure of infection! Nonetheless, we have removed the word indirect.
  5. Rows 101-102: This sentence is not so precise, I would remember to the authors how, according to all papers dedicated to the validation of Johne's disease diagnostic tools, the sensitivity is low, but strictly depending on the onset of the disease. Please reformulate. This has been adjusted.
  6. Rows 103-107: I would be very careful about this sentence, even if I have understood that the authors wanted to discuss that often is difficult/impossible discriminating between very low excreting MAP subject to false positive calls by PCR, the cycle of count (better say cycle of quantification according to the MIQE guidelines) are not absolute values and are extremely depending on the run, even in assay very well validated. For this reason, in order to check if the run was acceptable, negative and positive controls (these last should also have a confidence interval of acceptability) are included in each run. I would suggest to reformulate these sentences keeping in mind the above-mentioned concepts. The sentence has been reformulated (line 112-113).
  7. Row 148: Serum Sample. Information about the sera used in the longitudinal study is not present in this section,

The full supporting data for these samples is given in the Supplementary file.

…according to my suggestion I would remove this last part from the manuscript. They are given in lines 201-209, together with additional information in the Supplementary file. We would prefer to leave these additional 90 samples negative by MAP IDEXX in the paper, but have changed the way they are presented. We hope this will be acceptable to the referee.

  1. Rows 173-174: Please, did the herd introduce animals from farms with lower Johne's disease status? The Johne’s disease status of some of the APHA source farms is not known.

12  Rows 208: Please remove the space after 2.2. Done

  1. Row 237 : Please report here where or from who the lipids have been produced/purchased. Link back to section 2.2 has been included.
  2. Row 300:  Culture was not mentioned in this paper, please correct with PCR. Corrected
  3. Table 1 and elsewhere: why values are reported as median and not as mean? This is a generally accepted rule in the field of serological test? Because as mean the differences resulted less evident? Too much variability despite all the efforts to reduce it? I’m referring to what reported at rows 260-264. We used medians on the basis that medians are normally used for skewed number distributions, and in papers in human diagnostics we were told to replace means by medians. The medians are only used in the bar graphs, and do not affect significance values or sensitivity/specificity in RoC analysis.
  4. Table 1 is generally a little bit hard to follow, I suspect the authors have already tried different ways to show these data and this is the best they were able to obtain, so I don’t have so much suggestions to give to theme but the Table 1 still remains difficult to understand at first impact. I’m afraid we have made it more complicated by adding the statistical significances as requested by the second referee.
  5. Row 339: Please add a space after “only”. Done
  6. Row 352: Please check if the samples are positive to culture or PCR (as reported in row 156). Changed.
  7. 19. Rows 388-399: Please, for the above-mentioned reasons, remove this part. We presume the referee means lines 399-404 (the line numbers have changed from our original manuscript). We apologise that we way we presented this suggested this was a timeline of MAP disease development; in fact these are all MAP IDEXX and Bovigam negatives, and do not overlap with timeline studies of animals either artificially infected or in herds with high MAP incidence, which we do intend to submit for publication. The origin of the small number of elevated responses requires further study, but the fact that they are generally grouped suggests a change to the animals immune responses. WE would prefer to leave these negatives in the manuscript, and have changed the text in a way that we hope the referee will be happy with.
  8. Row 412: Please check if you are speaking about  culture or PCR. This has been adjusted.
  9. Row 408: I would suggest to reformulate taking into account the point discussed before about the cycles of quantification. Hopefully this is now OK.
  10. Row 442 and after: I cannot understand why animals housed inside should have much different environmental exposures than those housed outside. Please reformulate considering also what has been stated after. Same or different organisms. Stress The text has been changed, hopefully reflecting this comment.
  11. Row 471: Please add a “dot” at the end of the sentence. Done
  12. 24. Row 488: Please remove “early stages of infection” according to what already suggested above. For the same reason, remove from rows 532 to 545. We have changed this to ‘early stages of disease progression’. We can’t see the link to rows 532 to 545.
  13. Row 516: Please remove a dot and space. Done
  14. Row 561: Please add  “. Done
  15. Row 571: Please standardize characters (half in bold half not). Done

Round 2

Reviewer 1 Report

Comments and Suggestions for Authors

All my comments have been adressed. I thank the authors for the improving of the manuscript. 

Reviewer 2 Report

Comments and Suggestions for Authors

I appreciated the authors' additions and corrections. I like the paper in this form I think it is clearer and more complete.